# Towards Precision Sports Nutrition for Endurance Athletes: A Scoping Review of Application of Omics and Wearables Technologies

**DOI:** 10.3390/nu16223943

**Published:** 2024-11-19

**Authors:** Leon Bedrač, Leon Deutsch, Sanja Terzić, Matej Červek, Julij Šelb, Urška Ašič, Laure M. G. Verstraeten, Enej Kuščer, Filip Cvetko

**Affiliations:** 1The NU B.V., J.H. Oortweg 21, 2333 CH Leiden, The Netherlands; leon.deutsch@thenu.com (L.D.); sanja.terzic@thenu.com (S.T.); matej.cervek@thenu.com (M.Č.); julij.selb@thenu.com (J.Š.); urska.asic@thenu.com (U.A.); laure.verstraeten@thenu.com (L.M.G.V.); enej.kuscer@thenu.com (E.K.); filip.cvetko@thenu.com (F.C.); 2University Clinic of Respiratory and Allergic Diseases Golnik, Golnik 36, 4204 Golnik, Slovenia; 3Department of Human Movement Sciences, @AgeAmsterdam, Faculty of Behavioural and Movement Sciences Amsterdam Movement Sciences, Vrije Universiteit Amsterdam, 1081 HV Amsterdam, The Netherlands

**Keywords:** systems biology, nutrigenomics, endurance athletes, multi-omics, nutrition, continuous glucose monitoring, metagenomics

## Abstract

Background: Endurance athletes require tailored nutrition strategies to optimize performance, recovery, and training adaptations. While traditional sports nutrition guidelines provide a foundational framework, individual variability in metabolic responses underscores the need for precision nutrition, informed by genetic, biological, and environmental factors. This scoping review evaluates the application of systems biology-driven sports nutrition for endurance athletes, focusing on ‘omics’ and wearable technologies. Methods: A scoping review of the literature was conducted in PubMed, Scopus, and Web of Science in accordance with the PRISMA-ScR checklist. Research questions, search strategies, and eligibility criteria were guided by the Population–Concept–Context framework with the following inclusion criteria: original research in English, involving endurance athletes, systems biology approaches, and nutritional interventions or continuous glucose monitoring (CGM). Results: Fifty-two studies were included, with distance runners as the most studied cohort. Eleven studies used metagenomics, eleven CGM, ten nutrigenetics, ten metabolomics, seven multi-omics, one proteomics, one epigenomics, and one lipidomics. Over half (*n* = 31; 60%) were randomized controlled trials (RCTs) with generally high methodological quality. Conclusions: Most studies were proof-of-concept investigations aimed at assessing biomarkers; however, the evidence linking these biomarkers to performance, recovery, and long-term health outcomes in endurance athletes remains insufficient. Future research should focus on well-powered replicated crossover RCTs, multivariate N-of-1 clinical trials, 360-degree systems-wide approaches, and the validation of genetic impacts on nutritional interventions to refine dietary guidelines.

## 1. Introduction

Endurance athletes, such as triathletes, long-distance runners, race walkers, cyclists, swimmers, rowers, and cross-country skiers, represent the pinnacle of human athletic performance, characterized by a remarkable capacity for sustained physical exertion and exceptional metabolic efficiency [1]. Nutrition serves as a cornerstone of athletic success, fueling performance, promoting recovery, and facilitating adaptations to training demands [2]. Optimizing sports performance requires a tailored nutrition plan that considers individual athletes’ goals, needs, and physiological characteristics [3]. This is particularly important for endurance athletes, as the timing of nutrient intake, hydration, the digestibility of foods, and the absorptive capacity of nutrients are key factors for enhanced performance and recovery, in addition to body composition management [4]. Endurance athletes require different levels of carbohydrates, proteins, fats, and fluids based on body size, type of endurance sport, exercise intensity, and duration. Moreover, they have increased requirements for certain vitamins and minerals, particularly iron, calcium, and vitamin D. These needs, along with the necessity for antioxidants to mitigate oxidative stress from prolonged exercise, are sports-specific and vary upon the training goals [3,4]. While macronutrient-based dietary recommendations provide a foundational level of personalization, the substantial individual variability in the responses to exercise and nutrition highlights the need for a deeper understanding of metabolic heterogeneity and the physiological functions of nutrients [5,6]. Unlike most sports nutrition guidelines that rely on group averages and expert consensus, precision sports nutrition seeks to personalize nutritional practices for athletes to optimize performance and long-term health. To derive such recommendations, it is imperative to consider the multifaceted interactions between genetics, biological processes, nutrients, and environmental factors. A multimodal systems-wide approach is essential for integrating these complex relationships [5,6].

Systems biology integrates various high-throughput omics technologies and wearable sensor technologies to generate extensive datasets. By integrating these datasets through computational methods, researchers can construct predictive models that facilitate a deep understanding of individual responses to various interventions, including training and diet, on health, performance, and recovery [5,6]. While there is a lack of agreement on the definition of systems biology [7], for the purpose of this review, it is defined as an approach that incorporates at least one of the following ‘omics’ platforms or a combination thereof: genomics, transcriptomics, epigenomics, proteomics, metabolomics, and lipidomics for athlete characterization, along with metagenomics for microbiome analysis. Additionally, the inclusion of wearable sensor technology enhances this approach, as the integration of smart sensors and wearables represents a significant advancement in data collection methodologies by enabling the real-time continuous monitoring of physiological parameters [8].

Nutrigenomics and nutrigenetics examine the impact of genetic variations on individual metabolic responses to nutrients and the effect of dietary components on gene expression, utilizing genomic data and advanced genetic testing [9]. Transcriptomics investigates gene expression at the RNA level, providing insights into gene structure and function, which helps to elucidate molecular mechanisms governing biological processes [10]. Epigenomics, on the other hand, examines chemical modifications in chromatin, including folding, nuclear matrix attachment, nucleosome packaging, histone tail modifications, and DNA methylation, which influence gene expression without altering the DNA sequence [11]. Proteomics involves the large-scale analysis of protein structures, functions, and interactions, enabling the identification of biomarkers for health, disease processes, and therapeutic responses [12]. Metabolomics provides the ultimate molecular fingerprint of human physiology by comprehensively assessing small molecule metabolites within biological samples, thereby offering a robust framework for identifying metabolic shifts in response to diet, lifestyle, and environmental factors [13,14]. Lipidomics, a subfield of metabolomics, involves the analysis of cellular lipids using advanced mass spectrometry techniques to elucidate lipid metabolism and its biochemical pathways, providing insights into systemic metabolic perturbations [15]. Lastly, microbial metagenomics examines the role and modulation of the gut microbiome in the host’s nutrient uptake and energy metabolism [16]. Moreover, wearable technologies and real-time sensors, such as continuous glucose monitors, facilitate a more detailed understanding of how biological systems respond to various stimuli, thereby providing actionable data to inform training and nutrition strategies in real-time [17,18]. To harness the full benefits of these new technologies, it is essential to integrate systems biology data into biomedical models capable of generating actionable strategies. The goal is to identify ‘the right diet for the right person at the right time,’ rather than relying on a ‘one-diet-fits-all’ approach proposed by the general guidelines [5,19].

Despite the growing interest in precision sports nutrition [20], to our knowledge, no comprehensive review has synthesized the current application of omics and wearable sensor technologies in this field. The aim of this scoping review is to systematically evaluate the utility and applicability of systems biology-driven sports nutrition specifically focusing on endurance athletes. The primary objectives are to:Map the existing literature;Summarize key findings and insights;Identify research gaps to guide the field.

## 2. Materials and Methods

### 2.1. Research Methods

Steps for conducting this review were adopted from the five-step approach [21]. The review was conducted in accordance with the PRISMA extension for scoping reviews [22] (Appendix A). A scoping review methodology was selected based on the need to identify the type and extent of the research in the evolving field of precision sports nutrition and to identify research gaps. A comprehensive literature search was conducted in PubMed (MEDLINE), Scopus, and Web of Science between 1 June and 5 August 2024 and updated on 18 October 2024. Additionally, Google Scholar was used to assess gray area literature and to identify any missing articles. Citation searching, including reviewing reference lists of included studies and forward citation tracking, was also employed to ensure all relevant studies were captured. Search strings were developed and independently tested by two authors (L.B. and F.C.) to ensure the collection of relevant studies and to minimize any potential biases. Research questions, search strategies, and eligibility criteria were guided by the broad Population–Concept–Context (PCC) framework recommended by the Joanna Briggs Institute for scoping review [23]. The search terms encompassed three primary domains: (1) personalized and/or sports nutrition, (2) endurance athletes, and (3) systems biology approaches, including wearables technology. Detailed search strategies for each database can be found in the Appendix A.

### 2.2. Eligibility Criteria

To be included in this scoping review, peer-reviewed publications needed to: (1) involve endurance athletes, defined as athletes participating in sports requiring sustained physical effort such as marathon and long-distance running, race walking, cycling, triathlons, swimming, rowing, and cross-country skiing; (2) utilize at least one of the systems biology approaches (e.g., genomics, proteomics, metagenomics, metabolomics), including wearables technology in the context of sports nutrition; (3) investigate nutritional interventions and/or continuous glucose monitoring (CGM) in the context of endurance exercise; (4) examine the impacts of omics-driven nutritional approaches on health, performance, or recovery metrics; (5) be original research studies. Studies were excluded if they: (1) focused on non-endurance athletes; (2) did not report nutritional intervention and/or CGM data; (3) lacked the application of systems biology or ‘omics’ approaches. Additionally, the search was limited to articles published in the English language and restricted to human studies, with no restrictions on the publication period or study design. An updated search removed restrictions to human studies to increase the number of relevant studies. Finally, the studies had to be fully completed and published; abstract-only, presentation-only, and unpublished studies were excluded.

### 2.3. Study Selection

Two authors (L.B. and L.D.) independently screened titles and abstracts to identify potentially relevant studies. Full-text articles of the selected studies were retrieved and assessed for eligibility based on the inclusion criteria. Any disagreements on study selection or data extraction were resolved by consensus or after discussion with the third author (S.T.).

### 2.4. Data Charting

A standardized data extraction form was developed to systematically collect relevant information from the eligible studies. The data charting table was developed by the first author (L.B) and further improved and approved by the other authors to finally include key variables. Data extraction included study characteristics such as authors, publication year, and study design and population characteristics such as gender, age, training status, and performance level. Performance level was assessed by VO_2_max when reported (<65 mL kg^−1^ min^−1^ was used as a threshold for elite athletes), as well as other training status indicators such as weekly training volume and descriptors used in the original studies, including terms like ‘elite’, ‘sub-elite’, ‘non-elite’, ‘professional’, ‘amateur’, and ‘well-trained’. Intervention details were captured, including duration, frequency, and types of nutritional and exercise interventions, respectively. Additionally, information on the systems biology approach was extracted, focusing on the specific ‘omics’ platform and/or wearable technology used, along with the biological matrix employed for analysis. The outcomes of interests were related to performance, recovery, or health parameters. A thematic analysis was used to identify key findings and synthesize interpretation from the included studies. To assess the methodological quality of randomized controlled trials (RCTs), PEDro scale was employed [24] (Appendix A).

## 3. Results

### 3.1. Literature Search and Study Selection

A total of 1011 studies were identified through databases and registers, with three additional studies from Google Scholar, and 24 from citation searching. After removing duplicates and initial screening, 71 reports were sought for retrieval, of which 36 were excluded during eligibility assessment. Of the additional twenty-seven reports identified, twenty-six were assessed for eligibility and nine were excluded. In total, 52 studies were included in the review. The PRISMA flow diagram for study selection is presented in Figure 1.

### 3.2. Characteristics of Studies

Figure 2 illustrates the distribution of ‘omics’ platforms and wearable technologies employed in the studies included in this review. Ten out of fifty-two studies (19%) focused on nutrigenetics [26,27,28,29,30,31,32,33,34,35] (Table 1) and ten studies assessed metabolomics (19%) [36,37,38,39,40,41,42,43,44,45]. Additionally, one study each assessed proteomics (2%) [46], epigenomics (2%) [47], and lipidomics (2%) [48], while seven studies (13%) employed multi-omics approaches [49,50,51,52,53,54,55] (Table 2). Eleven studies (19%) investigated metagenomics [56,57,58,59,60,61,62,63,64,65,66] (Table 3), while CGM was used in eleven studies [67,68,69,70,71,72,73,74,75,76,77] (Table 4). No studies involving transcriptomics were identified. Overall, 31 (60%) of the studies were randomized controlled trials (RCTs), of which 20 (65%) were placebo-controlled and 19 (61%) adopted a crossover design [34,37,38,39,41,42,43,45,46,47,48,49,50,52,53,54,55,63,64]. Furthermore, eleven studies (21%) were non-randomized interventional studies [26,27,28,29,31,36,57,58,66,72,76], seven (13%) were longitudinal cohort studies [67,68,70,71,74,75,77], and three (6%) were case studies [61,69,73]. Study durations ranged from one day [70,73] to 12 months [75]. The studies were conducted between 2009 and 2024 across several continents, including North America (*n* = 19; 37%), South America (*n* = 5; 10%), Europe (*n* = 13; 25%), Asia (*n* = 8; 15%), and Australia (*n* = 7; 13%). A total of 1695 participants were included, of which 123 (7%) were classified as elite athletes. Sample sizes ranged from *n* = 1 [61,69,73] to *n* = 125 [28,29]. Most studies (*n* = 28; 54%) included both sexes albeit with a skew toward the male population. The majority focused on adult athletes, with seven studies (13%) including athletes under 18 years of age. Notably, 22 (42%) studies involved master athletes, defined as individuals over 40 years of age. Running was the most frequently studied endurance sport, accounting for 35% (*n* = 18) of the studies [26,27,28,29,31,40,44,52,56,60,62,63,68,70,72,73,77], followed by cycling (*n* = 15; 29%) [30,32,39,41,42,43,45,48,49,50,51,53,54,55,75]. Other studied disciplines included swimming (*n* = 1; 2%) [66], rowing (*n* = 1; 2%) [61], race walking (*n* = 3; 6%) [57,58,76], triathlons (*n* = 1; 2%) [59], and cross-country skiing (*n* = 1; 2%) [65]. Six studies (12%) reported mixed endurance type sports [33,35,37,38,67,71], and six studies (12%) did not specify the type of discipline [34,36,47,48,64,74] (Figure 2b). Moreover, two studies (4%) examined athletes with type 1 diabetes [67,75]. Of the interventional studies, 24 (46%) employed supplements as the nutritional intervention, with the remaining studies investigating individual foods or combinations thereof. The effect of ergogenic aids was assessed in 16 studies (31%).

### 3.3. Nutrigenetics

Table 1 details the application of nutrigenetics in precision sports nutrition for endurance athletes. The 10 studies reviewed assessed two types of supplementation interventions: five studies investigated the effect of genetic variations on caffeine metabolism, and the other five studies examined the impact of different gene variants related to antioxidant defense systems in response to pequi oil supplementation. Pequi oil showed anti-inflammatory properties and the potential to mitigate exercise-induced oxidative stress and cellular damage [29,78]. Significant associations were reported between genetic variations in endogenous antioxidant systems, specifically MnSOD, CAT, and GPX1 genes, and responses to pequi oil supplementation [27]. Notably, all studies evaluating pequi oil were conducted by the same research group, involving a similar sample size of recreational runners, suggesting the use of a potentially homogeneous cohort.

Two of the five caffeine-related nutrigenomics studies were also conducted by the same research group, likely using the same cohort [33,35]. Both studies employed placebo-controlled designs: the first study found that caffeine supplementation enhanced performance in CYP1A2 rs762551 ‘fast’ (AA) metabolizers [33], while the second study identified that athletes with the HTR2A CC genotype within this subgroup experienced the greatest ergogenic benefit [35]. In addition, a similar ergogenic effect of caffeine was observed in ‘fast’ (AA) metabolizers in a study of 35 trained male cyclists [30]. Furthermore, rs762551 ‘fast’ metabolizers showed improved cognitive performance in the psychomotor vigilance test [34]. However, the ADORA2A rs5751876 had no effect on cognitive performance, and neither the CYP1A2 rs762551 nor the ADORA2A rs5751876 influenced exercise performance. Pataky et al. [32] reported no significant ergogenic benefit from caffeine for rs762551 ‘fast’ (AA) metabolizers, though a small benefit was noted among rs762551 AC carriers, typically classified as ‘slow’ metabolizers.

**Table 1 nutrients-16-03943-t001:** Summaries of nutrigenetics studies’ characteristics and key findings in endurance athletes.

Authors, Publication Year	Study Design	Study Population	Analytical Platform; Matrix	Intervention	Key Findings
Miranda-Vilela et al., 2009 [26]	One-arm interventional design with a 14-day nutritional intervention between two running races; blood sample time points: after each race	124 recreational runners; 75 males and 49 females (age range: 15–67 years)	Genotyping (PCR-RFLP), target SNPs: MnSOD Val9Ala (rs1799725), CAT -21A/T (rs7943316), GPX1 Pro198Leu (rs1050450); blood	Volunteers participated in two races under identical training and environmental conditions, before (control) and after (treatment) 14 days of daily supplementation with 400 mg of pequi oil. Athletes chose the distance (4–21 km) based on their weekly training.	MnSOD Val/Ala heterozygotes had the least DNA and tissue damage, lowest lipid peroxidation, and best response to pequi oil against exercise-induced damage. No significant effects for CAT or GPX1 genes.
Miranda-Vilela et al., 2010 [27]	Same as [26]	119 recreational runners; 74 males and 45 females (age range: 15–67 years)	Genotyping (allele-specific PCR and PCR-RFLP), target polymorphisms: Hp, MnSOD Val9Ala (rs1799725), CAT -21A/T (rs7943316), GPX1 Pro198Leu (rs1050450); blood	Same as [26]	MnSOD Val/Val homozygotes, CAT A allele carriers, and GPX1 Pro allele carriers showed the best response to pequi oil for improving exercise-induced anisocytosis and blood oxygen-carrying capacity.
Miranda-Vilela et al., 2011 [28]	Same as [26]	125 recreational runners; 76 males and 49 females (age range: 15–67 years)	Genotyping (allele-specific PCR and PCR-RFLP), target polymorphisms: Hp, MnSOD Val9Ala, CAT -21A/T, GPx-1 Pro198Leu, GSTT1-null, ACE I/D, GSTM1-null, CK-MM TaqI, CK-MM NcoI, CRP G1059C, MTHFR C677T, MTHFR A1298C; blood	Same as [26]	Post-supplementation, Hp, ACE, GSTT1, and MTHFR A1298C affected lipid profile; MTHFR A1298C impacted CRP levels; and Hp and MnSOD influenced lipid peroxidation. In before–after comparisons, differences between ACE genotypes in leukogram and cholesterol, Hp and MnSOD in lipid peroxidation, and MTHFR A1298C in CRP disappeared.
Miranda-Vilela et al., 2016 [29]	Same as [26]	125 recreational runners; 76 males and 49 females (age range: 15–67 years)	Genotyping (allele-specific PCR), target SNP: IL-6 -174G/C (rs1800795); blood	Same as [26]	Pequi oil best protected against muscle damage in IL-6 GC genotypes; C allele carriers showed ↓ lipid peroxidation than GG homozygotes.
Womack et al., 2012 [30]	Randomized double-blind, placebo-controlled design with two separate time trials; blood sample time points: before each trial	35 trained male cyclists (mean age: 25.0 ± 7.3 years, VO_2_max: 59.35 ± 9.72 mL kg^−1^ min^−1^)	Genotyping (PCR-RFLP), target SNP: CYP1A2 rs762551; blood	Participants consumed caffeine (6 mg/kg) or placebo 60 min before each time trial. 40 km time trials were performed on an indoor cycle trainer on two separate mornings after a 12 h fast and at least 24 h without caffeine.	Caffeine ↓ average cycling time to a greater degree in CYP1A2 AA homozygotes (4.9%) compared to C allele carriers (1.8%).
Ribeiro et al., 2013 [31]	Same as [26]	123 recreational runners; 74 males and 49 females (age range: 15–58 years)	Genotyping (PCR-RFLP), target SNPs: ACTN3 R577X rs1815739, EPO rs1617640; blood	Same as [26]	Post-supplementation, the EPO TG genotype had ↓ CRP and GG had ↓ platelet count compared to TT; ACTN3 XX had ↓ MCH and ↑ lymphocyte count compared to RX. In before–after comparisons, ACTN3 RX showed ↓ AST, and XX showed ↓ CK.
Pataky et al., 2016 [32]	Randomized, counterbalanced, double-blind, placebo-controlled design with six cycling time trials, 3–7 days apart; blood sample time points: after the final time trial	38 recreational trained cyclists (min 1 day of cycling per week); 25 males and 13 females (mean age: 21 ± 1 years, VO_2_max: 51 ± 6 mL kg^−1^ min^−1^)	Genotyping (PCR-RFLP), target SNP: CYP1A2 rs762551; blood	Subjects ingested a 6 mg/kg caffeine or placebo capsule 1 h before each cycling trial. Each trial started with a 5 min warm-up with two mouth rinses, followed by a 3 km time trial. Subjects received two 25 mL mouth rinses with either 300 mg caffeine or a placebo. The four treatments were (1) placebo capsule + placebo rinse, (2) placebo capsule + caffeine rinse, (3) caffeine capsule + caffeine rinse, and (4) caffeine capsule + placebo rinse.	↑ power output in CYP1A2 AC heterozygotes by caffeine capsule (6.1%) and caffeine capsule + caffeine rinse (4.1%); AA homozygotes ↑ by 3.4% with the capsule + rinse but ↓ with the rinse alone (−0.4%); AC heterozygotes benefited more from caffeine capsule than AA genotypes.
Guest et al., 2018 [33]	Split-plot randomized, double-blinded, placebo-controlled design with three supplementation days, 1 week apart; saliva sample time point: before the intervention	101 competitive male athletes; CYP1A2 rs762551 AA genotype (mean age: 24 ± 4 years, VO_2_max: 49 ± 8 mL kg^−1^ min^−1^), AC genotype (age: 25 ± 5 years, VO_2_max: 47 ± 12 mL kg^−1^ min^−1^), CC genotype (age: 25 ± 5 years, VO_2_max: 44 ± 12 mL kg^−1^ min^−1^)	Genotyping (Sequenom MassArray platform, Sequenom Inc., San Diego, CA, USA), target SNP: CYP1A2 rs762551; saliva	Participants received placebo or caffeine (2 mg/kg or 4 mg/kg) before a 10 km cycling time trial.	Caffeine ↓ cycling time in the CYP1A2 AA genotype at both 2 mg/kg (−4.8%) and 4 mg/kg (−6.8%) without a dose difference; 4 mg/kg ↑ cycling time by 13.7% in the CC genotype; no effect in the AC genotype.
Carswell et al., 2020 [34]	Randomized double-blind, placebo-controlled crossover design with 3–9-day washout period between each supplementation day; blood sample time points: pre-supplementation, pre-exercise, and post-exercise	18 active adults (mean age: 24 ± 4 years); 12 males (VO_2_max: 49.5 ± 7.7 mL kg^−1^ min^−1^) and six females (VO_2_max: 43.2 ± 10.6 mL kg^−1^ min^−1^)	Genotyping (rhAmp assays), target SNPs: ADORA2A rs5751876, CYP1A2 rs762551; blood	Participants received caffeine (3 mg/kg) or placebo, with measures of endurance (15-min cycling, 70 min post-supplementation) and cognitive performance (pre-, 50-, and 95-min post-supplementation).	Caffeine ↑ performance similarly across CYP1A2 and ADORA2A genotypes. Faster reaction times and higher response speeds in CYP1A2 AA homozygotes with no differences in C-allele carriers or ADORA2A genotypes.
Guest et al., 2022 [35]	Whole-plot complete randomized block, double-blinded, placebo-controlled design with 3 supplementation days, 1 week apart; saliva sample time point: before the intervention	100 competitive male athletes; HTR2A rs6313 CC genotype (mean age: 24 ± 4 years, VO_2_max: 49 ± 8 mL kg^−1^ min^−1^), CT genotype (mean age: 25 ± 5 years, VO_2_max: 47 ± 12 mL kg^−1^ min^−1^), TT genotype (mean age: 25 ± 5 years, VO_2_max: 44 ± 12 mL kg^−1^ min^−1^)	Genotyping (Sequenom MassArray platform, Sequenom Inc., San Diego, CA, USA), target SNPs: HTR2A rs6313, CYP1A2 rs762551; saliva	Participants received placebo or caffeine (2 mg/kg or 4 mg/kg) before a 10 km cycling time trial.	4 mg/kg caffeine ↑ performance in individuals with both the HTR2A CC and CYP1A2 AA genotypes. Among CYP1A2 AA individuals, HTR2A CC genotypes outperformed T-allele carriers. No performance differences in CYP1A2 C allele carriers based on HTR2A genotype.

ACE: Angiotensin I-converting enzyme; ACTN3: α-actinin-3; ADORA2A: Adenosine A2A receptor; AST: aspartate aminotransferase; CAT: Catalase; CK: creatine kinase; CK-MM: Creatine kinase muscle type; CRP: C-reactive protein; CYP1A2: Cytochrome P450 1A2; EPO: Erythropoietin; GPX1: Glutathione peroxidase 1; GSTM1: Glutathione S-transferase M1; GSTT1: Glutathione S-transferase T1; Hp: haptoglobin; HTR2A: 5-Hydroxytryptamine receptor 2A; IL-6: Interleukin-6; MCH: Mean corpuscular hemoglobin; MnSOD: Manganese superoxide dismutase; MTHFR: Methylenetetrahydrofolate reductase; PCR: Polymerase chain reaction; PCR-RFLP: Polymerase chain reaction-restriction fragment length polymorphism; SNP: single-nucleotide polymorphism.

### 3.4. Metabolomics, Proteomics, Epigenomics, Lipidomics, and Multi-Omics

Table 2 gives an overview of the application of various ‘omics’ platforms in nutritional interventions targeting endurance athletes. Of the 20 studies reviewed, twelve were led by Nieman et al., involving various cohorts of athletes, primarily cyclists and runners. This research group has been instrumental in advancing the application of ‘omics’ technologies in sports nutrition, notably with their 2012 study comparing banana consumption with isotonic carbohydrate supplementation during a 75 km cycling trial [39]. Their research has primarily focused on investigating the impact of polyphenols from fruits, supplements, and nuts on exercise-induced immune system perturbations and recovery [40,41,42,48,49,50,51,52,53,54,55]. Using a metabolomics-based approach, they showed that both blueberry and banana consumption attenuated exercise-induced increases in pro-inflammatory oxylipins in cyclists, with significant inter-individual variability in gut-derived phenolic metabolite levels [5,51]. Earlier research revealed distinct phenolic metabolite profiles in runners following polyphenol-rich soy protein supplementation, albeit with no attenuation of traditional inflammation markers [40]. Similarly, a more recent study found that 2-week mango fruit supplementation did not reduce post-exercise inflammatory response compared to water ingestion [54]. In contrast, banana consumption consistently outperformed carbohydrate-only ingestion by enhancing anti-inflammatory responses and improving monocyte metabolism [48,49]. Nieman et al. [52] also examined the effects of astaxanthin supplementation using multi-omics, and while astaxanthin failed to suppress exercise-induced cytokine and oxylipin elevations, it countered reductions in plasma immunoglobulins, suggesting an immune-normalizing effect post-exercise. In addition, Nieman et al. [53] assessed a 2-week nitrate supplementation regimen from beet-based products, observing increases in anti-inflammatory oxylipins and reduced complement activation despite no reduction in pro-inflammatory oxylipins. However, the acute effects of nitrates on metabolic recovery warrant further investigation, as Stander et al. [44] reported no accelerated recovery with beetroot juice following a marathon. Furthermore, a recent multi-omics study by Nieman et al. [55] on cranberry beverage supplementation in cyclists revealed an upregulation of innate immune system proteins, contrary to what was measured with beet supplementation without alteration of gut microbiome diversity.

Research by other groups using ‘omics’ approaches has highlighted the benefits of carbohydrate–protein blends [36] and oligopeptides [45] for improving metabolic recovery and lipid metabolism, while post-exercise protein supplementation has been shown to mitigate disruptions in sulfur amino acid metabolism [43]. Conversely, proteomics research indicates that low energy availability (LEA) impairs performance and immune function in female endurance athletes [46]. Finally, a single epigenomics study suggests that dietary fat and exercise modulate muscle DNA methylation, evidenced by significant hypomethylation in CpG islands [47].

**Table 2 nutrients-16-03943-t002:** Summaries of proteomics, metabolomics, epigenomics, and multi-omics studies’ characteristics and key findings in endurance athletes.

Authors, Publication Year	Study Design	Study Population	Analytical Platform; Matrix	Intervention	Key Findings
Chorell et al., 2009 [36]	Non-randomized controlled trial of four interventions; blood sample time points: before and post-exercise (0 h, 0.25 h, 0.5 h, 1 h, 1.5 h)	24 non-elite male athletes (age: 25.7 ± 2.7 years, VO_2_max: 59.1 ± 7.3 mL kg^−1^ min^−1^)	Predictive metabolomics (GC−TOF MS, HMCR); plasma	Participants ingested one of four beverages after 90 min of cycling across four test sessions: LCHO (1 g CHO/kg), HCHO (1.5 g CHO/kg), LCHO-P (1 g/kg CHO, 0.5 g PROT/kg), or water; PROT included 90% casein and 10% whey protein; CHO included 37.5% maltodextrin, 31.25% sucrose, 15.6% glucose, and 15.6% galactose.	LCHO-P: ↑ amino acids, PSU, cholesterol, and 4-deoxyerythronic acid; ↓ 3-methylhistidine; water: ↑ fatty acids; LCHO and HCHO: ↑ sugar levels; PSU ↑ with LCHO-P, suggesting ↑ protein synthesis; ↑ adenine catabolism and metabolic stress in high VO_2_max individuals (↑ uric acid levels).
Nelson et al., 2012 [37]; Nelson et al., 2013 [38]	Randomized, double-blind, placebo-controlled, crossover design with a 14-day washout period; blood sample time points: before and post-exercise (0 h, 0.5 h, 1 h, 1.5 h, 2 h, 3 h) on days 1 and 6	12 well-trained male cyclists or triathletes (mean age, 35 ± 10 years; VO_2_max: 64.8 ± 6.8 mL kg^−1^ min^−1^)	Targeted metabolomics (GC–MS); plasma	Athletes ingested either LEUPRO (protein/leucine/carbohydrate/fat: 20/7.5/89/22 g/h) or CON (carbohydrate/fat: 119/22 g/h) for 1–3 h post-exercise during 6 days of high-intensity training.	LEUPRO altered amino acid and acylcarnitine metabolism (↓ muscle damage). No significant performance improvements [37]. LEUPRO ↑ neutrophil oxidative burst after 6 days of training. Acutely, LEUPRO ↓ neutrophil oxidative burst (↑ myristic acid levels) [38].
Nieman et al., 2012 [39]	Randomized, crossover design with a 3-week washout period; blood sample time points: before and post-exercise (0 h, 1 h)	14 trained non-elite male cyclists (mean age 37.0 ± 7.1 years; VO_2_max: 58.6 ± 5.2 mL kg^−1^ min^−1^)	Untargeted metabolomics (GC–MS); plasma	Subjects ingested 0.4 g/kg carbohydrate from bananas (BAN) or from a standard 6% CHO beverage (Gatorade™, Chicago, IL, USA) before exercise and 0.2 g/kg body weight every 15 min during the 75 km time trials.	No significant differences between groups in blood glucose and performance metrics. Exercise ↑ levels of multiple inflammatory and oxidative stress markers with different patterns for IL-10 and IL-8 between CHO and BAN and FRAP in BAN. Differences in dopamine levels between groups.
Nieman et al., 2013 [40]	Randomized, double-blind, placebo-controlled, parallel group design of 17 days supplementation period with 3-day periods of exercise test inserted at day 14; blood sample time points: before- and after 14-day supplementation, and immediately and 14 h after third day of running	31 non-elite competitive long-distance runners: 18 males and 13 females, (mean age: 33.7 ± 6.8 years; VO_2_max: 50.0–64.4 mL kg^−1^ min^−1^)	Untargeted metabolomics (UHPLC/MS/MS2, GC–MS); plasma	2 × 20 g daily of 3:1 blueberry–green tea–polyphenol soy protein complex over a 17-day period, including a 14-day pre-exercise period, and during each day of the 3-day intensified exercise period (2.5 h treadmill running at ~70% VO_2_max).	Exercise: significant physiological, inflammatory, and oxidative stress. Supplementation: no ↓ stress biomarkers post-exercise, ↑ gut-derived phenolic metabolites. Exercise-induced gut permeability led to ↑ fat oxidation and ketogenesis in recovery.
Nieman et al., 2014 [41]	Randomized, crossover design with 2-week supplementation period followed by a time trial and a 2-week washout period; blood sample time points: 45 min before and post-exercise (0 h, 1.5 h, 21 h)	19 non-elite male competitive cyclists (mean age: 38.0 ± 1.6 years; VO_2_max: 51.7 ± 1.4 mL kg^−1^ min^−1^)	Untargeted metabolomics (UHPLC/MS/MS, GC–MS); plasma	2 weeks of pistachio (3 oz/day) or no pistachio supplementation followed by 75 km time trial after an overnight fast. Participants also consumed 1.5 oz before and after the 1-h time trial.	Pistachio ↓ time trial performance by 4.8%. Exercise induced changes in inflammatory, oxidative stress, and metabolic markers; ↑ raffinose correlated with oxidative stress markers; ↑ specific bile acids, amino acids, fatty acid metabolites, and lysolipids.
Nieman et al., 2015 [42]	Randomized, crossover design with a 2-week washout period; blood sample time points: before and post-exercise (0 h, 1.5 h, 21 h)	20 non-elite male competitive cyclists (mean age: 39.2 ± 1.9 years; VO_2_max: 51.0 ± 1.4 mL kg^−1^ min^−1^)	Metabolomics (UPLC–MS/MS); plasma	Participants completed three 75 km cycling time trials under three conditions: water only, bananas and water, and pears and water. CHO intake (0.4 g/kg pre-exercise, 0.15 g/kg every 15 min) was provided for banana and pear groups.	Banana and pear: ↑ cycling performance (5.0% and 3.3%), compared to water; ↓ cortisol, IL-10, and total leukocytes; ↑ blood glucose, insulin, and FRAP; Banana: ↑ fructose, dopamine, serotonin-related metabolites, and antioxidant markers (pear showed similar but less pronounced effects). Pear consumption associated with gastrointestinal discomfort.
Olsen et al., 2020 [43]	Double-blind, randomized, crossover design with at least a 6-day washout period between two experimental interventions; blood sample time points: day 1 post-exercise (0 h, 0.25 h, 0.5 h, 1 h, 1.5 h, 2 h); day 2 before exercise, during exercise (15 min, 30 min, and 70 min after the start of the time trial), and 15 min post-exercise exercise	Eight elite male cyclists (mean age: 22.7 ± 3.5 years; VO_2_max: 74.7 ± 4.01 mL kg^−1^ min^−1^)	Targeted metabolomics (LC–MS/MS); plasma and urine	Athletes cycled to exhaustion and received supplementation immediately after exercise and at 30 min intervals for 120 min: CHO+PROT: 0.8 g/kg/h CHO (glucose + maltodextrin, 1:1) and 0.4 g/kg/h PROT (whey); CHO: 1.2 g/kg/h (glucose + maltodextrin, 1:1). After an ~18 h recovery period, athletes completed a 60 min time trial	The CHO+ PROT group cycled 8.5% faster than the CHO group. Post-exercise: methionine ↓ by 55% in CHO vs. 33% in CHO + PROT (*p* < 0.001). The methionine/homocysteine ratio ↓ by 54% in CHO vs. 27% in CHO+PROT (*p* < 0.001). Cystathionine ↑ by 72% in CHO vs. 282% in CHO + PROT. Total cysteine, taurine, and glutathione ↑ by 12%, 85%, and 17%, (during exercise).
Stander et al., 2021 [44]	Randomized, placebo-controlled, participant groups were matched according to predicted marathon finishing times; blood sample time points: before the race and post-exercise (0 h, 24 h, 48 h)	31 marathon athletes; 19 males and 12 females; placebo group (mean age: 39 ± 12 years, marathon finishing time 04:30:25 ± 00:36:48), beetroot group (mean age: 42 ± 10 years, marathon finishing time 04:07:08 ± 00:39:16)	Untargeted metabolomics (GC-GC-TOFMS); plasma	During the two consecutive days following the race, athletes received either beetroot juice or isocaloric placebo. Supplements were consumed as follows: 3 × 250 mL on marathon day (immediately after, ±3 h post-race, and at 20:00); 3 × 250 mL the day after (upon waking, with lunch, and supper); 250 mL upon waking on the second day post-marathon.	Both the beetroot and placebo groups returned to pre-marathon levels in metabolic profiles within 48 h. Random interindividual variation observed post-marathon in two metabolites deriving from CHO (arabitol and xylose) and two from odd-chain fatty acids (nonanoate and undecanoate). No immediate metabolic recovery benefits were identified.
Jin et al., 2023 [45]	Randomized, controlled, single-blinded, crossover trial with a 2-week washout period; blood sample time points: fasted, before and post-exercise (90 min after first exercise, after time trial), fasted after 19 h recovery	16 male cyclists (mean age: 17.0 ± 1.0 years; VO_2_max: 56.3 ± 5.8 mL kg^−1^ min^−1^)	Quantitative metabolomics (UPLC–MS/MS); plasma	Athletes consumed two 6% CHO and electrolyte beverages, with or without 2.7% FOPS, across two test sessions involving intraday fasting, 30 min of sitting still, 85 min of prolonged exercise, a 5 min sprint, a 60 min recovery period, a 20 min time trial, and recovery until the next morning. FOPS provided 35 g of oligopeptides, including 7.5 g of essential amino acids and 1.5 g of leucine per athlete during the trial.	101 TGs, 32 FAAs and their metabolites, and eight Krebs cycle metabolites were identified; five of twenty plasma FAAs ↑ 20 min after oligopeptide ingestion before exercise. Serum TGs and non-esterified fatty acids were ↓ in the experimental group post-exercise and post-time trial; ↓ plasma TGs post-exercise and during fasting in the experimental group, ↑ fat oxidation.
Jeppesen et al., 2024 [46]	Randomized, single-blinded crossover study with a 14-day dietary intervention followed by 3 days of refueling and an 11-day washout period; blood sample time points: before, 7 days into and after 14 days of both interventions, and after each 3-day refueling period	12 female endurance athletes (mean age: 26.8 ± 3.4 years; VO_2_max: 55.2 ± 5.1 mL kg^−1^ min^−1^)	Proteomics (Olink Proteomics, Uppsala, Sweden, Target 96 Inflammation Panel); plasma	Participants completed two 14-day dietary phases: OEA (50 kcal/kg FFM/day) and LEA (22 kcal/kg FFM/day). After each phase, a 3-day OEA refueling period was implemented, with phases separated by an 11-day washout. Eight 20 min cycling time trials were performed: before the intervention, on day 7, after 14 days of OEA and LEA, and following each 3-day refueling period.	LEA ↑ NADPH oxidase and systemic cortisol, altered inflammatory proteins, and ↑ exercise-induced hydrogen peroxide emission in peripheral blood mononuclear cells. Performance ↓ after LEA with limited recovery post-refueling and impaired immune function; 78/96 plasma proteins quantifiable; LEA ↓ 5 and ↑ 2 proteins.
Gorski et al., 2023 [47]	Randomized, counterbalanced, cross-over design with a 1–2-week washout period; blood sample time points: before morning exercise, during exercise, and post-recovery; drink after morning exercise (0 h, 0.5 h, 1 h, 2 h, 3 h)	Nine well-trained male athletes; (mean age: 30 ± 7 years; VO_2_max: 66 ± 6 mL kg^−1^ min^−1^)	Epigenomics (Infinium Methylation EPIC BeadChip Array, Illumina, San Diego, CA, USA); plasma	Standardized diet for 24 h before the lab visit: 40 kcal/kg FFM (1.2 g/kg FFM fat, 6.0 g/kg FFM CHO, 1.35 g/kg FFM PROT). Post-exercise day 1: EB-HF: 30 kcal/kg FFM (73% fat, 16% CHO, 11% PROT); ED-LF: 9 kcal/kg FFM (10% fat, 53% CHO, 37% PROT). Day 2: Both groups consumed a recovery drink (1.2 g/kg FFM CHO and 0.38 g/kg FFM protein) 30 min post-morning exercise.	Baseline: EB-HF showed hypermethylated DNA (60%) compared to ED-LF. Post-exercise: EB-HF: significant hypomethylation in regulatory regions (CpG islands) and ↑ expression of HDAC2, MECR, IGF2, and c13orf16; ED-LF: ↑ expression of HDAC11; EB-HF: epigenetic and transcriptional changes that support exercise recovery and metabolism.
Nieman et al., 2019 [48]	Randomized, crossover, counterbalanced four-arm design with a 2-week washout period; blood sample time points: before and post-exercise (0 h, 0.75 h, 1.5 h, 3 h, 4.5 h, 24 h, 45 h)	20 non-elite competitive cyclists: 14 males (mean age: 37.1 ± 2.5 years; VO_2_max: 47.0 ± 1.5 kg^−1^ min^−1^); six females (mean age: 43.7 ± 2.2 years; VO_2_max: 46.5 ± 2.8 mL kg^−1^ min^−1^)	Lipidomics (LC–MRM-MS); plasma	Overnight-fasted cyclists completed a 75 km time trial while ingesting either water (3 mL/kg), a 6% sugar beverage (0.2 g/kg CHO), Cavendish bananas (0.2 g/kg carbohydrate), or polyphenol-rich mini-yellow bananas (0.2 g/kg carbohydrate) every 15 min of exercise.	CHO intake ↓ ARA and DHA mobilization, and CYP-derived oxylipin generation after 75 km cycling. Oxylipin levels ↑ in the water trial, while CHO ↓ this rise, particularly for nine of twelve CYP-derived oxylipins. This effect was most pronounced in the first three hours of recovery, with most oxylipins coming from ARA, including over 15 eicosanoids from LOX and CYP pathways.
Nieman et al., 2018a [49]	Randomized, crossover, counterbalanced four-arm design with a 2-week washout period; blood sample time points: before and post-exercise (0 h, 0.75 h, 1.5 h, 3 h, 4.5 h, 24 h, 45 h)	20 non-elite competitive cyclists: 14 males (mean age: 37.1 ± 2.5 years; VO_2_max: 47.0 ± 1.5 kg^−1^ min^−1^); six females (mean age: 43.7 ± 2.2 years; VO_2_max: 46.5 ± 2.8 mL kg^−1^ min^−1^)	Multi-omics: global metabolomics (UPLC–MS/MS)/lipidomics (LC–MRM-MS); plasma	Overnight-fasted cyclists completed a 75 km time trial while ingesting either water (3 mL/kg), a 6% sugar beverage (0.2 g/kg CHO), Cavendish bananas (0.2 g/kg carbohydrate), or polyphenol-rich mini-yellow bananas (0.2 g/kg carbohydrate) every 15 min of exercise.	CHO from bananas or sugar beverages ↓ exercise-induced stress responses (cortisol, inflammation, and lipid disturbances). Water-only group: 109 metabolites ↑ >2-fold, while 71 ↓ by >0.5-fold. Post-exercise: 65% of the ↓ metabolites were triacylglycerol esters; ↑ metabolic disruption in the water-only condition compared to the banana and sugar beverages. Banana: ↓ COX-2 mRNA expression in monocytes.
Nieman et al., 2018b [50]	Randomized, double-blind, placebo-controlled, crossover design	59 participants from three studies [41,42,49]	Multi-omics: global metabolomics (UPLC–MS/MS)/lipidomics (LC–MRM-MS); plasma	Overnight-fasted participants were subjected to different nutritional interventions during a 75 km cycling time trial. Two trials: 3 mL/kg of water or water containing 0.15–0.20 g/kg of CHO every 15 min (CHO sources: bananas, pears, or a 6% sugar beverage). One trial: 3 oz of pistachio nuts per day for 2 weeks. On the day of the trial, they ingested 1.5 oz of pistachio nuts before and after the 1-h time trial.	26 key metabolites associated with exercise-induced changes; CHO ingestion ↓ the metabolic impact of exercise by 28–47% compared to water-only, depending on CHO type and recovery time.
Nieman et al., 2020 [51]	Randomized, double-blind, placebo-controlled, parallel four group design with 2-week supplementation period followed by time trial and 2.5 days of recovery monitoring; blood sample time points: before and after supplementation, and post-exercise (0 h, 1.5 h, 3 h, 5 h, 24 h, 48 h)	59 non-elite competitive cyclists; 40 males and 19 females (age range: 36–41 years; VO_2_max: 44.1–52.3 mL kg^−1^ min^−1^)	Multi-omics: metabolomics (UPLC–MS/MS)/lipidomics (LC–MRM-MS); plasma	Freeze-dried blueberry ingestion (26 g/d) vs. placebo for 2 weeks. Both groups were further randomized to ingestion of a water-only control or water with a CHO source (Cavendish bananas, 0.2 g/kg CHO every 15 min) during a 75 km cycling time trial.	Exercise ↑ 64 of 67 oxylipins; both blueberry and banana intake ↓ pro-inflammatory oxylipins within first 3 h of recovery. Blueberry intake ↑ 24 gut-derived phenolics and ↓ post-exercise oxylipins, while acute banana intake strongly ↓ 10 pro-inflammatory oxylipins.
Nieman et al., 2023 [52]	Randomized, double-blind, placebo-controlled, crossover design with two 4-week supplementation periods and a 2-week washout period; blood sample time points: before and after supplementation, and post-exercise (0 h, 1.5 h, 3 h, 24 h)	18 recreational distance runners; 11 males (mean age: 40.7 ± 2.7 years; VO_2_max: 52.7 ± 2.9 mL kg^−1^ min^−1^) and seven females (mean age: 43.7 ± 2.9 years; VO_2_max: 52.7 ± 2.9 mL kg^−1^ min^−1^)	Multi-omics: untargeted proteomics (MS-DIA)/targeted oxylipins profiling (LC–MRM-MS); plasma	4 weeks of 8 mg astaxanthin supplementation prior to 2.25 h treadmill running test.	No effect on exercise-induced muscle soreness and muscle damage, and no elevation in six plasma cytokines and 42 oxylipins. Supplementation countered exercise-induced ↓ in 82 plasma proteins related to immune functions (restoration of IgM). Significant between-subject variability observed in 500 identified plasma proteins.
Nieman et al., 2024a [53]	Randomized, double-blind, placebo-controlled, crossover design with two 2-week supplementation periods and a 2-week washout period; blood sample time points: before and after supplementation, and post-exercise (0 h, 1.5 h, 3 h, 24 h)	20 non-elite recreational cyclists; 14 males (mean age: 46.5 ± 2.6 years; VO_2_max: 41.2 ± 1.6 mL kg^−1^ min^−1^) and six females (mean age: 51.3 ± 4.3 years; VO_2_max: 40.9 ± 2.9 mL kg^−1^ min^−1^)	Multi-omics: untargeted proteomics (MS-DIA)/targeted oxylipins profiling (LC–MRM-MS); plasma	2 weeks of BEET supplementation prior to a 2.25 h cycling test in a fasted state. The BEET supplement contained 212 mg of nitrates; 200 mg caffeine; 44 mg vitamin C; 40% RDA of thiamine, riboflavin, niacin, and vitamin B6; and 2.5 g of a mushroom blend (*Cordyceps sinensis* and *Inonotus obliquus*).	Cycling ↑ 41 of 67 oxylipins, with BEET supplementation further ↑ two anti-inflammatory oxylipins (18-HEPE, 4-HDoHE); BEET impacted 66 of 616 proteins, ↓ 45 and ↑ 21 compared to placebo; BEET ↓ inflammation-related proteins, involved in complement activation, acute phase response, and immune function.
Sakaguchi et al., 2024 [54]	Randomized, crossover design with a 2-week washout period; blood and urine sample time points: pre- and post- supplementation and blood sampling post-exercise (0 h, 1.5 h, 3 h, 24 h)	22 cyclists; 13 males (mean age: 43.2 ± 2.1 years; VO_2_max: 43.4 ± 2.3 mL kg^−1^ min^−1^) and nine females (mean age: 37.9 ± 3.2 years; VO_2_max: 37.9 ± 2.9 mL kg^−1^ min^−1^)	Multi-omics: targeted lipidomics (LC–MRM-MS)/metabolomics (UPLC–ESI-TOF); plasma and urine	Cyclists ingested 330 g of mango/day with 0.5 L water or 0.5 L of water alone for 2 weeks, followed by a 2.25 h cycling bout challenge; 1.5 h after exercise, mango group consumed 165 g of mango, while water-only group drank 0.45 L of a 6% CHO sports drink.	After supplementation, mango-derived phenolic metabolites ↑; no effect on post-exercise oxylipin patterns or inflammation; significant post-exercise ↑ in 49 oxylipins and inflammation. Mango supplementation did not alter these responses compared to water.
Nieman et al., 2024b [55]	Randomized, double-blind, placebo-controlled, crossover design with two 4-week supplementation periods and a 2-week washout period; blood sample time points: before and after supplementation, and post-exercise (0 h, 1.5 h, 3 h, 24 h), stool and urine samples collected pre- and post supplementation.	25 non-elite cyclists; 17 males (mean age: 43.2 ± 2.2 years; VO_2_max: 46.2 ± 2.1 mL kg^−1^ min^−1^) and eight females (mean age: 41.8 ± 2.9 years; VO_2_max: 37.4 ± 2.1 mL kg^−1^ min^−1^)	Multi-omics: untargeted proteomics (MS-DIA)/targeted oxylipins profiling (LC–MRM-MS)/untargeted metabolomics (UHPLC–HRMS)/metagenomics (WGS; plasma, urine and stool	In random order, the cyclists supplemented their diets with 240 mL/d of cranberry or a 240 mL/d placebo beverage for 4 weeks, followed by the 2.25 h cycling challenge. The cranberry beverage included 317 ± 19 mg of polyphenols, 294 ± 26 mg proanthocyanidins, 41 ± 5 mg anthocyanins, and 9 g of intrinsic sugars per 240 mL serving.	Cycling ↑ 53 of 75 oxylipins in both groups,); 595 plasma proteins detected with two clusters differing between both groups. Proteins related to innate immunity ↑, whereas proteins related to platelets degranulation ↓ in supplementation group; 5719 taxa identified via WGS with no genus or species-level differences between supplementation and placebo group.

18-HEPE: 18-hydroxyeicosapentaenoic acid; 4-HDoHE: 4-hydroxy-docosahexanoic acid; ARA: arachidonic acid; BCAAs: branched-chain amino acids; CFU: colony forming units; CHO: carbohydrate; CYP: cytochrome P450; DHA: docosahexaenoic acid; EB-HF: energy balance high-fat; ED-LF: energy deficit low fat; FAAs: free amino acids; FFM: fat-free mass; FOPS: food-derived oligopeptides; FRAP: ferric reducing ability of plasma; GC–MS: gas chromatography coupled with mass spectrometry; GC–MS: gas chromatography–mass spectrometry; GC−TOF MS: gas chromatography/time-of-flight mass spectrometry; GCxGC-TOFMS: two-dimensional gas chromatography time-of-flight mass spectrometry; HCHO: high-carbohydrate; HMCR: hierarchical multivariate curve resolution; IL-10: interleukin 10; IL-6: interleukin 6; IL-8: interleukin 8; LCHO: low-carbohydrate; LCHO-P: low-carbohydrate-protein; LC–MRM-MS: liquid chromatography coupled with multiple reaction monitoring mass spectrometry; LC–MS/MS: liquid chromatography coupled with tandem mass spectrometry; LC–MS: liquid chromatography coupled with mass spectrometry; LEA: low energy availability; LOX: lipoxygenase; MS-DIA: Mass spectrometry with data-independent acquisition; OEA: optimal energy availability; PROT: protein; PSU: pseudouridine; PUFAs: polyunsaturated fatty acids; RDA: recommended dietary allowance; SCFAs: short-chain fatty acids; TGs: triglycerides; UHPLC–HRMS: ultra-high performance liquid chromatography coupled with high resolution mass spectrometry; UHPLC/MS/MS: ultra-high performance liquid chromatography coupled with tandem mass spectrometry; UPLC: ultra-performance liquid chromatography; UPLC–ESI-TOF: ultra-performance liquid chromatography electrospray ionization time of flight mass spectrometry; WGS: whole genome shotgun sequencing.

### 3.5. Metagenomics

Table 3 provides a summary of metagenomic studies investigating the role of the microbiome in endurance athletes, focusing on its impact on health, performance, and recovery. Seven of the eleven studies examined the effects of prebiotic or probiotic supplementation, while three explored the influence of various dietary strategies. Most studies reported non-significant differences in alpha diversity following nutritional/probiotic interventions or physical activity. Murtaza et al. [57] found negative correlations between *Bacteroides* and *Dorea* abundances and performance measures following the consumption of the low-carbohydrate high-fat (LCHF) diet in elite race walkers. In a separate analysis of the oral microbiome of the same cohort, the LCHF diet had the most pronounced impact, reducing the relative abundance of *Haemophilus*, *Neisseria*, and *Prevotella*, and was linked to increased nitrate/nitrite-reducing activity and nitric oxide generation [58].

Studies investigating the effects of probiotic supplementation on exercise performance revealed a positive impact of *Bifidobacterium longum* subsp. *longum* OLP-01, *Veillonella atypica* FB0054, *Lactobacillus plantarum* PS128, and *Bifidobacterium animalis* subsp. *lactis* BL-99 supplementation on endurance performance in middle- and long-distance runners, recreational athletes, triathletes, and cross-country skiers, respectively [59,60,64,65]. Additionally, OLP-01 supplementation protected against muscle mass reduction [60], whereas BL-99 supplementation improved lipid metabolism, evidenced by reductions in triglycerides and low-density lipoprotein cholesterol [65]. Finally, Jaago et al. [61] demonstrated that prebiotic supplementation with dietary fibers induced a shift from acetate- to butyrate-producing bacteria, promoting beneficial gut microbiome modulation and a significant decrease in alpha diversity.

Further investigations explored the effects of cocoa flavanols and a FODMAP diet, both examining the link between the gut microbiome and intestinal permeability (‘leaky gut’) in runners. Tabone et al. [62] measured serum markers of intestinal permeability, including intestinal fatty acid-binding protein (I-FABP) and lipopolysaccharides (LPS). Similarly, Gaskell et al. [63] assessed intestinal permeability by measuring plasma microbial DNA levels and short chain fatty acids (SCFA) in response to low- and high-FODMAP diets during prolonged exercise under heat stress. Both studies reported increased intestinal permeability in response to exercise; however, no significant changes were observed between or within groups, regardless of the nutritional intervention.

**Table 3 nutrients-16-03943-t003:** Summaries of metagenomics studies characteristics and key findings in endurance athletes.

Authors, Publication Year	Study Design	Study Population	Analytical Platform; Matrix	Intervention	Key Findings
Moreno-Pérez et al., 2018 [56]	Randomized, double-blind, placebo-controlled pilot design with a 10-week supplementation period; stool sample time points: before and after the intervention	18 male cross-country runners who regularly engaged in endurance training (240 min/week); PRO group (mean age: 34.90 ± 9.49 years), CHO group (mean age: 35.38 ± 9.00 years)	Amplicon metagenomic sequencing (MiSeq platform, llumina, San Diego, CA, USA) and targeted metabolomics (GC–MS); stool	The PRO group received a blend of whey isolate (10 g) and beef hydrolysate (10 g) daily for 10 weeks. The control (CHO) group received maltodextrin	PRO group: ↓ *Bifidobacterium longum*, *Roseburia*, *Blautia*, *Coprococcus*; PRO vs. CHO groups: ↑ *Bacteroides*, ↓ *Citrobacter* and *Klebsiella*; no significant differences in fecal SCFA levels.
Murtaza, et al., 2019a [57], Murtaza et al., 2019b [58]	Non-randomized, controlled design with three dietary intervention groups during a 3-week period of intensified training; stool sample time points: before and after intervention	21 male elite race walkers (age range: 20–35 years); HCHO group (VO_2_max: 61.6 ± 6.8 mL kg^−1^ min^−1^), PCHO group (VO_2_max: 64.6 ± 5.3 mL kg^−1^ min^−1^), LCHF group (VO_2_max: 66.3 ± 4.8 mL kg^−1^ min^−1^)	Amplicon metagenomic sequencing (MiSeq platform, Illumina, San Diego, CA, USA); stool [57] and saliva [58]	HCHO diet: 60% carbohydrate (~8.5 g/kg/day), 16% protein (~2.1 g/kg/day), 20% fat; PCHO diet: similar macronutrient composition as HCHO but periodized in consumption across the day and throughout the week; LCHF diet: 78% fat, 17% protein (~2.2 g/kg/day), 3.5% carbohydrate (<50 g/day)	PCHO diet: ↑ Ruminococcaceae, *Coprococcus*, *Bifidobacterium*, *Streptococcus,* and *Akkermansia muciniphila*; ↓ *Bilophila*; HCHO diet: ↑ Clostridiaceae, Lachnospiraceae, Ruminococcaceae, ↓ *Sutterella*; LCHF diet: ↑ *Dorea*, *Bacteroides*, *Akkermansia*, ↑ *Faecalibacterium*, *Veillonella*, *Streptococcus*, *Succinivibrio*, *Odoribacter*, *Lachnospira*, *Bifidobacterium* [57]; LCHF diet: ↑ Gram-positive bacteria (*Streptococcus*, *Faecalibacterium*, *Peptostreptococcus*, *Rothia*), ↓ Gram-negative bacteria (*Neisseria* and *Prevotella*); HCHO/PCHO diets: ↑ Gram-negative bacteria (*Haemophilus*, *Leptotrichia*) [58].
Huang et al., 2020 [59]	Randomized, double-blind, placebo-controlled design with a 4-week probiotic supplementation period; stool sample time points: after intervention	20 male triathletes; L. plantarum group (mean age: 21.6 ± 1.3 years, VO_2_max: 55.5 ± 8.6 mL kg^−1^ min^−1^), placebo group (mean age: 21.9 ± 1.4 years, VO_2_max: 56.6 ± 9.0 mL kg^−1^ min^−1^)	Amplicon metagenomic sequencing (Roche 454 GS FLX, LabX, Ontario, ON, Canada); stool	Daily supplementation of *Lactobacillus plantarum* PS128 at a dose of 3 × 10^10^ CFU for 4 weeks	PS128 group: ↓ *Anaerotruncus*, *Caproiciproducens*, *Coprobacillus*, *Desulfovibrio*, *Dielma*, *Holdemania* and *Oxalobacter*, ↑ *Akkermansia*, *Bifidobacterium*, *Butyricimonas* and *Lactobacillus*.
Lin et al., 2020 [60]	Randomized, double-blind, placebo-controlled design with a 5-week probiotic supplementation period (3 weeks training, 2 weeks de-training). Stool sample time points: before and after intervention	21 well-trained middle- and long-distance runners; 14 males and seven females (age range: 20–30 years)	Amplicon metagenomic sequencing (HiSeq2500 platform, llumina, San Diego, CA, USA); stool	Daily supplementation of *Bifidobacterium longum* subsp. *longum* OLP-01 at a dose of 1.5 × 10^10^ CFU for 5 weeks. 12 min Cooper’s running test was conducted before and after the supplementation period; distance traveled was recorded every 3 min (thirdrd, sixth, ninth, and twelfth min)	The OLP-01 group: ↑ *Bifidobacterium* genus and the specific probiotic strain *Bifidobacterium longum* subsp. *longum*.
Jaago et al., 2021 [61]	Case study of an athlete over an 8-month period with a 30-day supplementation period; stool sample time points: preseason, at week 27, and at week 31 after 30 days of supplementation.	18-year-old male academic rower	Amplicon metagenomic sequencing (MiSeq platform, llumina, San Diego, CA, USA); stool	Daily supplementation with 20 g of prebiotic mix containing 8.79 g dietary fiber, consisting of resistant starch (2.25 g), arabinoxylan (2.05 g), citrus fiber (2 g), beta-glucans (1.03 g), inulin (1.03 g), and rye fiber (0.57 g) for 30 days	↓ Firmicutes/Bacteroidetes ratio during period of intense competition and ↑ after fiber consumption.
Tabone et al., 2022 [62]	Randomized, placebo-controlled design with a 10-week intervention period and two exercise bouts; stool sample time points: pre- and post-exercise session; blood sample time points: pre- and post-exercise bout (T1, T2, T3, T4)	42 male cross-country athletes (mean age: 36.5 ± 9.0 years; VO_2_max: 59.7 ± 5.1 mL kg^−1^ min^−1^)	Amplicon metagenomic sequencing (MiSeq Illumina, San Diego, CA, USA) and targeted and untargeted metabolomics (LC–HRMS); stool and blood	Supplementation with 5 g of fat-reduced cococa (425 mg flavonols, CO group) or placebo; exercise test pre- and post- supplementation (10 min treadmill warm-up at 60% max HR, run at 1% slope at 10 km/h until exhaustion); training session during 10-week supplementation (5–6 x/week)	Serum: ↑ I-FABP (intestinal permeability) after exercise in both groups, no change in LPS; no change in metabolic profiles after consumption; four metabolites differed in T3 CO/T4 CO group. Stool: no significant changes in gut microbiota post-supplementation; two polyphenol metabolites differed between CO and placebo post-supplementation, with no change in metabolic profiles.
Gaskell et al., 2023 [63]	Randomized, double-blind, crossover design with 24 h diet intervention and 1-week washout period; stool sample time points: pre-EHS; blood sample time ponts: pre- and post-EHS	13 non-heat acclimatized recreationally competiteve endurance and ultra-endurance runners with a history of Ex-GIS; eight males and five females (mean age: 34 ± 7 years; VO_2_max: 63.9 ± 9.7 mL kg^−1^ min^−1^)	Amplicon metagenomic sequencing (MiSeq Illumina, San Diego, CA, USA) and targeted metabolomics (GC); stool and blood	HFOD (energy 2762 ± 844 kcal/day, PRO 104 ± 32 g/day, CHO 409 ± 139 g/day, fat 70 ± 19 g/day, fibre 54 ± 17 g/day, total FODMAP 51 ± 30 g/day) or LFOD diet (energy 2503 ± 640 kcal/day, protein 92 ± 26 g/day, carbohydrate 352 ± 94 g/day, fat 68 ± 17 g/day, fibre 50 ± 11 g/day, total FODMAP 2 ± 1 g/day) 24 h prior exercise (2 h run at 60% VO_2_max in T_max_ 35.6 °C, 22.6% RH)	Plasma: ↑ microbial DNA post-EHS in LFOD and HFOD; ↑ *Delftia* and ↓ *Serratia* post-EHS (LFOD); ↑ *Bacillus* post-exercise (HFOD); ↑ total plasma SCFA and acetate in HFOD vs. LFOD (pre-EHS). Stool: ↑ Ruminococcaceae, ↑ Firmicutes and ↓ Bacteroidota (LFOD vs. HFOD); ↑ total plasma SCFA, acetate, propionate and butyrate in HFOD vs. LFOD (pre-EHS)
Gross et al., 2023 [64]	Randomized, double-blind, placebo-controlled, crossover design with two 2-week supplementation periods and a 3-week washout period; stool sample time points: before and after each supplementation period	Seven recreational athletes; three males and four females (age: 30.7 ± 7.5 years; VO_2_max: 49.2 ± 8.4 mL kg^−1^ min^−1^)	Shotgun metagenomic sequencing (NovaSeq 6000, llumina, San Diego, CA, USA), untargeted metabolomics (UHPLC–HRMS); stool	Daily supplementation of *Veillonella atypica* FB0054 at a dose of 1 × 10^10^ CFU for 14 days. Treadmill time to exhaustion run test was conducted before and after each supplementation period	No changes in specific taxa or functions observed after placebo use, the washout, or FB0054 use. 14 metabolites differed significantly between the FB0054 use and both baseline and placebo.
Li et al., 2023a [65]	Randomized, single-blind, placebo-controlled design with an 8-week probiotic supplementation period; stool sample time points: before and after intervention	16 male national top-level cross-country skiers; control group (mean age: 19.3 ± 0.7 years, VO_2_max: 55.9 ± 4.4 mL kg^−1^ min^−1^), probiotic group (mean age: 19.6 ± 1.1 years, VO_2_max: 55.8 ± 5.4 mL kg^−1^ min^−1^)	Shotgun metagenomic sequencing (sequencing platform using a high-intensity DNA nanochip technique) and untargeted metabolomics (LC–MS); stool	Yoghurt with the addition of 1 × 10^9^ CFU of *Bifidobacterium animalis* subsp. *lactis* BL-99, four times per day for 8 weeks. VO_2_max and isokinetic muscle strength test were assessed before and after the intervention	40-fold ↑ of *B. animalis* in the BL-99 group, 2-fold ↑ in the placebo group; BL-99 combined with training improved lipid metabolism (↓ TGs and LDL) and ↑ VO_2_max and knee extensor strength); BL-99 ↑ DHA, adrenic, linoleic, and acetic acids, and ↓ glycocholic and glycodeoxycholic acids.
Li et al., 2023b [66]	One-arm interventional study with a 5-day probiotic supplementation period; stool sample time points: on a regular training day and post-intervention	15 elite open-water swimmers; eight males (mean age: 18.32 ± 4.41 years) and seven females (mean age: 18.04 ± 2.96 years)	Amplicon metagenomic sequencing (NovaSeq 6000, Illumina, San Diego, CA, USA) and untargeted metabolomics (HPLC–HRMS); stool	2 g of probiotic formula (inulin, oligofructose, lactitol, solid sterilized fermented carrot juice, *Bifidobacterium lactis* HN019, *Lactobacillus acidophilus* NCFM, *Lactobacillus plantarum* Lp-115, *Bifidobacterium longum* Bl-05) were administered twice daily over five training days	Female athletes: ↓ Firmicutes (positive correlation with organic acids and derivatives). *Pusillimonas*, *Acinetobacter*, *Aeromonas*, and *Stenotrophomonas* (Proteobacteria) associated with phenylpropanoids, polyketides, organic oxygen compounds, and organic acids. Male athletes: Proteobacteria (positive correlation with organosulfur compounds), bacteroidetes (negative correlation with alkaloids), *Lactobacillus* (Firmicutes) (positive association with organophosphorus compounds). Athletes ↓ pathways related to endocrine resistance, sphingolipid metabolism, and estrogen signaling.

CFU: colony forming units; CHO: carbohydrate; EHS: exertional-heat stress; DHA: docosahexaenoic acid; Ex-GIS: exercise-associated gastrointestinal symptoms; FODMAP: fermentable oligo- di- mono-saccharides and polyols; GC: gas chromatography; GC–MS: gas chromatography coupled with mass spectrometry; HCHO: high carbohydrate; HFOD: high fermentable oligo- and mono-saccharide; HPLC–HRMS: high-performance liquid chromatography coupled with high-resolution mass spectrometry; HR: heart rate; I-FABP: intestinal fatty acid-binding protein; LDL: low-density lipoprotein; LCHF: low-carbohydrate high-fat; LC–MS: liquid chromatography coupled with mass spectrometry; LFOD: low fermentable oligo- and mono-saccharide; LPS: lipopolysacharyde; ML: machine learning; PCHO: periodized carbohydrate; PRO: protein; RH: relative humidity; SCFA: short chain fatty acid; TG: triglycerides; UHPLC–HRMS: ultra-high performance liquid chromatography coupled with high-resolution mass spectrometry.

### 3.6. Wearables: Continuous Glucose Monitoring

Table 4 summarizes studies investigating the use of CGM technology during endurance events, encompassing both healthy individuals and those with diabetes. Most studies were observational, focusing on the impact of physiologically relevant parameters during and after exercise interventions. Notably, three studies compared CGM measurements from interstitial fluids with glucose concentrations from venous or capillary blood [71,72,77]. In elite race walkers, glucose levels were comparable to those of the general population, despite variations in diet and activity levels, with males exhibiting higher mean 24 h glucose levels than females [76]. Ultra-endurance runners generally maintained normoglycemia during extreme events; however, the risk of hypoglycemia increased at peak exertion [68,69,73,77]. Conversely, male athletes with type 1 diabetes (T1D) demonstrated a decline in glucose levels throughout a cycling race but experienced hyperglycemia post-race [67]. Another study conducted on a 12 professional cyclists with T1D highlights the unique glycemic challenges this cohort face, particularly managing hyperglycemia during competition and nocturnal hypoglycemia after non-competitive exercise [75]. Similarly, the risk of hyperglycemia increased in the 48 h following cycling and ultra-endurance trail events in healthy athletes, particularly after carbohydrate intake, with elevated basal glucose levels persisting for up to five days [71,77]. Finally, higher carbohydrate consumption before endurance events and increased energy intake during events were associated with improved performance [68,69,70], while glucose oxidation was impaired during submaximal exercise in overreached athletes [74].

**Table 4 nutrients-16-03943-t004:** Summaries of studies characteristics and key findings employing continuous glucose monitoring in endurance athletes.

Authors, Publication Year	Study Design	Study Population	Analytical Platform; Matrix	Intervention	Key Findings
Yardley et al., 2015 [67]	Prospective observational cohort during an endurance cycling race; interstitial glucose measured continuously from the day before the race, through the race, and overnight after the race	Six male athletes with T1D (mean age: 36.3 ± 9.3 years)	CGM (various devices); Interstitial fluid	Participants were asked to maintain their regular prerace routines for insulin dosage and food intake and to record their food intake before, during, and after the race.	Three participants experienced mild to moderate hypoglycemia during the event; all experienced hyperglycemia 3 h post-exercise; ↓ insulin administration pre-race and 40–60 g/h of CHOs ↓ the occurrence of hypoglycemia and avoided hyperglycemia during the race.
Ishihara et al., 2020 [68]	Prospective observational cohort study during a 160 km ultra-marathon race; interstitial glucose measured continuously during the race	Seven experienced ultramarathon runners; four males (mean age: 41.5 ± 6.2 years), three females (mean age: 42.6 ± 1.2 years)	CGM (FreeStyle Libre, Abbott Diabetes Care, Alameda, CA, USA); Interstitial fluid	Participants were asked to record their food and drink intake throughout the race. Running time and speed for each of 11 race segments were also collected.	Runners consuming <0.8 g/kg/h of CHOs had ↓ speed; the lowest and average glucose ↑ from resting levels correlated positively with running speed.
Ishihara et al., 2021 [69]	Case study during a 438 km ultramarathon; interstitial glucose measured continuously from 1 day before the race, during the 7-day ultra-marathon, and 3 days after	44-year-old female professional trail runner	CGM (FreeStyle Libre, Abbott Diabetes Care, Alameda, CA, USA); Interstitial fluid	Participant’s food and drink intake was recorded by accompanying runners. Running speed was measured via 33 timing gates.	Minimal diurnal glucose fluctuations and slight total glucose ↑ during the ultramarathon (limited sleep); glucose levels were not associated with running pace; ↑ pace was associated with ↑ nutrient and solid food intake.
Kinrade and Galloway, 2021 [70]	Prospective observational cohort study during a competitive 24 h event; interstitial glucose measured continuously during the race.	18 amateur ultra-endurance runners; 11 males (mean age: 39.3 ± 4.1 years, VO_2_max: 52.0 ± 5.1 mL kg^−1^ min^−1^) and seven females (mean age: 45.0 ± 4.7 years, VO_2_max: 47.1 ± 7.2 1 mL kg^−1^ min^−1^)	CGM (FreeStyle Libre, Abbott Diabetes Care, Alameda, CA, USA); Interstitial fluid	Dietary intake for 48 h pre-race and during the race was recorded using a weighed food intake method.	No association between mean interstitial glucose and dietary intake, or with race distance; runners who consumed ≥40 g/h CHO covered a greater distance compared to <40 g/h.
Kulawiec et al., 2021 [71]	Prospective observational cohort study with a monitoring and testing period; interstitial glucose measured continuously during the endurance test	10 sub-elite athletes; seven males and three females (age range: 22–50 years, VO_2_max: 37–67 mL kg^−1^ min^−1^)	CGM (Ipro2, Medtronic Minimed, Northridge, CA, USA; Guardian Real-time device, Medtronic Minimed, Northridge, CA, USA; Optium Xceed, Abbott Diabetes Care, Alameda, CA); Interstitial fluid	Glucose levels, exercise, and nutrition were monitored for 4–6 days. Athletes performed an endurance exercise test to exhaustion after 1–2 days of monitoring.	Glycemic variability and response to CHO intake ↑ on the testing day and normalized the next day; overnight glucose levels remained ↑ up to 3–4 days post-test.
Clavel et al., 2022 [72]	Prospective interventional study assessing the validity of CGM against finger prick measures in 4 days over 2 weeks; interstitial glucose measured every 10 min, finger-prick blood glucose measured over four different periods (post-breakfast, pre-exercise, exercise, and post-exercise).	Eight recreational athletes regularly participating in running and resistance-based training (8 ± 2 h per week); five males and three females (mean age 30.8 ± 9.5 years)	CGM (FreeStyle Libre, Abbott, France) and finger prick measures (FreeStyle Optium, Abbott, France); Interstitial fluid and capillary blood	Two breakfasts were provided before exercise: CHO (65 ± 7 g of carbohydrates, 9 ± 1 g of proteins and 1 ± 0 g of fat, 311 ± 31 kcal) and PROT (1 ± 0 g of carbohydrates, 30 ± 0 g of proteins and 23 ± 0 g of, 311 ± 31 kcal). The exercise routine included a 10 min low-intensity run, high-intensity intervals, and a 10 min walk.	The CGM device is accurate at rest but not reliable during exercise, especially when CHOs are consumed beforehand.
Takayama and Mori, 2022 [73]	Case study during a 24 h marathon; interstitial glucose measured continuously during the race	32-year-old male ultra-marathon runner (VO_2_max: 67.6 mL kg^−1^ min^−1^)	CGM (FreeStyle Libre, Abbott Diabetes Care, Alameda, CA, USA); Interstitial fluid	Nutrition intake during the week leading up to the 24 h ultramarathon was recorded using MyFitnessPal App (MyFitnessPal, Inc., San Francisco, CA, USA). Speed was calculated from laps per hour.	Glucose levels remained stable during the race due to adequate CHO intake prior and during the race. No significant correlation with running speed.
Coates et al., 2023 [74]	Prospective observational cohort study during 5-week training block	11 endurance athletes; eight males and three females (mean age: 28 ± 6 years, VO_2_max: 56.5 ± 7.3 mL kg^−1^ min^−1^)	CGM (Supersapiens, Atlanta, GA, USA; Abbott Libre Sense, Abbott Park, IL, USA); Interstitial fluid	Glucose levels monitored during 5-week training block (1 week reduced training, 3 weeks high- intensity overload, 1 week recovery). After each block, a cycling test and 5 km time-trial were conducted, followed by 50 g glucose ingestion and glucose levels recorded each minute starting 15 min after ingestion.	Glucose levels and carbohydrate oxidation ↓ during submaximal cycling test, but not 5 km time-trial after high-intensity overlaod week; CGMs during submaximal exercise following standardized nutrition could be employed as a monitoring tool to detect overreaching in endurance athletes.
van Weenen et al., 2023 [75]	Prospective observational cohort study during an entire competitive season, including during competitive- (CE) and non-competitive exercise (NCE)	12 professional male cyclists with type 1 diabetes (mean age: 25.6 ± 4.4 years, VO_2_max: 70.6 ± 4.0 mL kg^−1^ min^−1^)	CGM (Dexcom G6, Dexcom, San Diego, CA, USA); Interstitial fluid	Participants were monitored the entire competitive season. Duration, intensity factor, variability index, heart rate/power zones were also measured during exercise.	Time spent in hypoglycemia was ↓ in CE vs, NCE. Time in hyperglycemia ↑ in CE vs. NCE; In CE: time in range ↓ to 60.4 ± 13.0%, time in hyperglycemia ↑ (38.5 ± 12.9%), hypoglycemias not significant in CE phase (1.1 ± 1.4%).
Bowler et al., 2024 [76]	Non-randomized, controlled design with two 4-day trial periods, separated by three days; interstitial glucose measured continuously during the two trials	12 elite race walkers; seven males and five females (mean age: 22.5 ± 3.5 years, VO_2_max: 61.6 ± 7.3 mL kg^−1^ min^−1^)	CGM (Freestyle Libre 2, Abbott Diabetes Care, Alameda, CA, USA); Interstitial fluid	Participants were provided a standardized diet (225 kJ/kg/day, 8.5 g/kg/day CHO, 2.1 g/kg/day protein, 1.2 g/kg/day fat) during each trial. Exercise routine included steady state race-walk on day 1, economy and biomechanical testing on day 2, resistance training on day 3, and a 10 km race walk on day 4.	Glycemic variability in athletes was comparable to healthy individuals and ↓ than T2D, despite a high-CHO diet and intense training. Males had ↑ 24 h mean glucose levels than females, even with standardized diet and exercise.
Parent et al., 2024 [77]	Prospective observational cohort study during a 156 km ultra-trail race; interstitial glucose measured continuously from a day before the race until 10 days after	55 ultra-endurance runners; 34 males and seven females (mean age: 43.7 ± 9.6 years)	CGM (Freestyle Libre Pro IQ, Abbott, Alameda, CA, USA); Interstitial fluid	Food intake data were collected the day before, during, and the day after the race.	No major glycemic events occurred during the race. Significant hyperglycemia risk was observed during recovery, up to 48 h. Glycemic metrics did not affect performance or behavioral alertness.

CGM: continuous glucose monitoring; CHO: carbohydrate; PROT: protein; T1D: type 1 diabetes; T2D: type 2 diabetes.

## 4. Discussion

This scoping review represents the first systematic overview of various ‘omics’ and CGM wearable technology approaches within the emerging field of precision sports nutrition, focusing on endurance athletes. The objective was to map the existing literature, present key findings, and identify research gaps to guide the field. A total of 52 studies of varying quality met the inclusion criteria. Furthermore, RCTs examining metabolomics and multi-omics approaches demonstrated high methodological quality as assessed by the PEDro scale, while RCTs in other areas achieved a moderate quality score (Appendix A). With the exception of nutrigenetics studies assessing the ergogenic effects of caffeine, most were proof-of-concept investigations focused on biomarkers associated with health, performance, and metabolic recovery, primarily derived from plasma or stool samples. Running and cycling were the predominant sports disciplines, likely due to the feasibility of obtaining real-time data from blood samples in controlled laboratory environments. These sports offer robust models for elucidating exercise-induced biomolecular pathways through metabolomics, proteomics, lipidomics, and multi-omics approaches, which can be applied in situ. Conversely, ultra-endurance events, such as ultramarathons and prolonged cycling, primarily employed CGM technology to capture dynamic glycemic responses in relation to prolonged physical exertion and nutritional interventions. Additionally, endurance athletes in weight-bearing disciplines (e.g., race walkers, triathletes, and runners) provide relevant models for investigating intestinal perturbations and gut microbiota composition in response to dietary macronutrient manipulations, as well as targeted prebiotic and probiotic supplementation. Our findings indicate a paucity of evidence linking the identified biomarkers to performance, recovery, and long-term health outcomes in endurance athletes. Additionally, inter-individual variability in measured plasma biomarkers in responses to exercise and nutritional interventions was observed [44,51,52]. Moreover, many studies were limited by small sample sizes and inadequate reporting of training status. Furthermore, the majority of studies recruited heterogeneous groups of athletes across various ages and sexes, with no reported stratification of results based on these demographics. This is an important consideration, as it may interfere with the interpretation of individual responses to dietary interventions based on genotype, metabolite profile, and enterotype.

### 4.1. Nutrigenetics

An individual’s genome typically differs from the reference human genome at approximately 4–5 million sites [79]. Such genetic variation impacts a wide range of biological processes, including those related to nutrient metabolism and the body’s response to these nutrients. Tailoring nutrition to an athlete’s DNA could potentially enhance performance and optimize health outcomes [80]. Numerous gene–nutrient interactions have been proposed to impact sports performance. The data, however, are not robust, and the effects of genetic variations on performance need to be validated to develop precision nutrition strategies [81]. Despite the growing popularity and availability of genetic testing, only ten studies in this review investigated nutrigenetics in endurance athletes, focusing on only two supplements.

Pequi oil, derived from the pulp of the pequi fruit, is a carotenoid-rich supplement that remains underexplored in sports nutrition. Research by Miranda-Vilela et al. has reported associations between genetic variations in antioxidant systems and responses to pequi oil supplementation [26,27,28,29,31]. However, these studies have several methodological limitations, including before-and-after designs and inadequate statistical adjustments for multiple associations. As a result, the reported associations may be due to chance and need validation in independent cohorts.

Conversely, caffeine is one of the most extensively studied compounds in nutrigenetics, with significant inter-individual variability in its effects on sports performance linked to genetic factors [81]. Over 95% of caffeine is metabolized by the CYP1A2 enzyme [33]. The -163A>C (rs762551) SNP has been shown to influence CYP1A2 inducibility and activity, enabling the stratification of individuals as ‘slow’ (AC, CC genotypes) or ‘fast’ (AA genotype) caffeine metabolizers [33]. Another significant gene in caffeine metabolism and response is ADORA2A; the rs5751876 SNP is used to categorize individuals based on their caffeine sensitivity as ‘high’ (TT genotype) or ‘low’ (CT or CC genotypes) [34]. While placebo-controlled studies suggest that caffeine benefits ‘fast’ metabolizers, with enhanced performance and cognitive function, the findings are inconsistent. For instance, some studies reported minimal or no significant ergogenic benefits in ‘fast’ metabolizers [32]. These inconsistencies warrant further research in larger, independent athlete cohorts to refine recommendations on optimal dosing, timing, and exercise conditions for caffeine supplementation.

### 4.2. Proteomics, Metabolomics, Epigenomics, Lipidomics, and Multi-Omics

While genomics holds promise for identifying athletic talent, optimizing performance, predicting sports-related injury risks, and determining recovery timelines, the athlete’s genome represents only one aspect of sports science [82]. The physiological underpinnings of athletes are shaped by the complex interplay of genetic traits and a host of environmental stimuli necessitating advanced approaches such as metadata analysis and multiple omics profiling to be fully elucidated [83,84,85]. The emerging field of multi-omics facilitates a comprehensive evaluation of metabolic responses to exercise, nutrition, and lifestyle interventions [86]. Technological advancements have revolutionized exercise science, enabling the simultaneous measurement of thousands of molecules from minimal quantities of biological fluids, cells, and tissues. While traditional metabolic studies of intense exertion focused on a limited set of biochemical markers, recent research has demonstrated that acute vigorous exercise alters approximately 15,000 transcripts, over 300 proteins, and more than 700 metabolites and lipids [86,87].

Endurance athletes encounter considerable physiological stress, leading to muscle damage, oxidative stress, inflammation, gastrointestinal disturbances, and transient immune suppression [39,41,48,49,50,51]. In contrast to moderate exercise, which generally enhances immune function, this immunosuppression may persist for hours to days and is frequently associated with an increased incidence of upper respiratory tract infections 1–2 weeks post-competition. Such infections can disrupt training goals within meso- and macrocycles, thereby affecting peak performance. Exercise-induced inflammation activates immune cells, which can be monitored in real-time using metabolomics and lipidomics [51,52,53,54,55]. Intense exercise elevates oxylipin production and lipid mobilization, driving post-exercise inflammation and oxidative stress. Both acute carbohydrate intake and chronic polyphenol consumption attenuate, though do not fully suppress, these responses [51]. These findings provide valuable insights into the complex role of post-exercise inflammation and oxidative stress, revealing their dual function in driving essential adaptations at optimal levels, while excessive responses may compromise recovery and long-term health [88].

While carbohydrate supplementation is well-established for modulating immune responses to prolonged exercise, the use of polyphenols remains an emerging area [5]. Peri-workout carbohydrate intake effectively reduces post-exercise stress hormones, inflammation, and fatty acid mobilization. Fruits rich in carbohydrates and polyphenols offer immune-modulating benefits comparable to sports drinks, with the added potential for long-term health benefits. Notably, individual responses to different polyphenol-rich carbohydrate sources vary, with one study showing up to a 14-fold difference in plasma levels of blueberry-derived metabolites among cyclists. Higher responders exhibited lower post-exercise plasma oxylipin levels, highlighting the need for further research to elucidate these variations [5,51]. Additionally, the reduction of post-exercise plasma oxylipin levels appears to be influenced by the specific macronutrient and polyphenol composition of the fruit consumed, suggesting that the type of polyphenols and macronutrient balance as well as dosing regimen play a role in modulating exercise-induced inflammatory responses [54,55]. Finally, the limited research on modulation of the proteome and epigenome induced by specific nutritional interventions in endurance athletes underscores an important area for future investigation.

### 4.3. Metagenomics

Over the past two decades, the human gut microbiome has been recognized as a crucial component of health, influenced by various factors such as diet, medications, and physical activity. Research on the gut microbiota in athletes faces several confounding factors. These include the timing of sample collection (in-season versus off-season), the specific sport discipline (as different exercises impact the microbiota differently), and methodological constraints such as small sample sizes that can lead to variability in results [16,89]. Exercise and dietary fiber consumption are associated with increased bacterial diversity and the presence of SCFA-producing bacterial species, like butyrate. Butyrate is the primary energy source for colonocytes and has anti-inflammatory properties. Increased butyrate production may be crucial for mitigating mucosal inflammation (leaky gut), a key concern for endurance athletes, who experience substantial gastrointestinal stress during intense training and competition [61,89]. Further research into biomarkers indicative of intestinal epithelial injury may elucidate the role of the microbiome and its contribution to the development of leaky gut. Available evidence suggests that leaky gut may serve as an adaptive response to exercise-induced physiological stressors in athletes, leading to alterations in gut microbiota composition, which underscores the importance of considering both factors in understanding this condition [62,63]. While emerging research indicates promising potential of harnessing the gut microbiome to enhance endurance performance, further investigation is required to establish causal relationships between microbiome shifts induced by diet and exercise, and their impacts on health and athletic performance. Additionally, the influence of peri-workout simple carbohydrate intake on dental health and its association with microbial shifts warrants investigation in the field of metagenomics.

### 4.4. Wearables: Continuous Glucose Monitoring

Energy availability (EA) is crucial for athletic performance, and its effective management is essential for endurance athletes due to increased energy demands over extended periods. Blood glucose levels serve as a key indicator of EA, influenced by diet and exercise, and exhibit significant inter-individual variability [17]. CGM, originally developed for diabetes management, is increasingly used in endurance sports to evaluate glucose variability during exercise and recovery. This technology allows athletes to monitor glucose levels and responses to food intake and exercise [17,90]. Associations between glucose levels during exercise and performance remain inconclusive [70,73], although one study found a positive correlation between moderate glucose increases and running speeds [68]. Moreover, while CGM data generally align with capillary and venous blood glucose measurements [71,77], some studies report increased measurement bias during exercise [72]. CGM technology offers valuable insights into glucose fluctuations and the real-time EA of athletes [17]; however, the extent to which interstitial glucose values reflect physiological disturbances from daily training and fueling or represent an athlete’s adaptive response to training remains to be fully elucidated. Recently established reference ranges for the glycemic variability provide a more robust framework for interpreting CGM data in endurance athletes [76]. Finally, while only CGM met the inclusion criteria for this scoping review within the concept of systems biology in endurance athletes, other wearable technologies and real-time sensors show promise for developing precision sports nutrition strategies and warrant further investigation [18].

### 4.5. Limitations, Knowledge Gaps, and Future Directions

The main limitation of this scoping review is the exclusion of non-published data in the rapidly evolving field of systems biology, which may have restricted the identification of relevant studies. Despite the promise of a systems biology framework to address the complex interactions that elicit an athlete’s metabolic response to nutrition-derived stimuli, significant knowledge gaps persist. These gaps are largely due to the intricate interactions within the biological contexts and the limitations of current bioinformatics tools in interpreting large datasets. Moreover, assessing the impact of dietary nutrient alterations on athletes’ metabolic profiles is further complicated by interactions with non-nutrient signals from environmental exposure [5].

Advancing the application of systems biology in precision sports nutrition for endurance athletes requires a deeper understanding of the underlying mechanisms governing metabolic heterogeneity in response to nutrition and exercise. To achieve this, a comprehensive 360-degree systems biology approach is needed, incorporating the collection of individual-specific data, identification of biomarkers, or a distinct ‘fingerprint’ of biomarkers associated with performance, health, and recovery, as well as the development of predictive biomedical models. This approach should generate evidence-based nutritional strategies while continuously monitoring health and performance outcomes. To drive progress in the field, addressing the following research gaps is warranted:Study design: Implement adequately powered RCTs employing replicated crossover designs to elucidate the sources of variability in responses to nutritional interventions. These studies should focus on the diet-by-person interaction by analyzing within-person variance [91]. In addition, a multivariate N-of-1 and aggregated N-of-1 clinical trial should be conducted to assess individual responses [92].In situ research: Bridge the gap between laboratory findings and practical applications by conducting exercise interventions that accurately resemble the physiological demands of endurance sports. Ensure that nutritional strategies, particularly those implemented during exercise, are standardized, feasible, and applicable in real-world settings.CGM: Conduct studies with larger cohorts and clearly defined dietary protocols to elucidate the relationships between diet, glucose levels, and variability, and their effects on athletic performance, recovery, and health.Metagenomics: Initiate large-scale, multi-center shotgun sequencing studies to elucidate the microbiome’s role in athletic performance. Although cost-prohibitive, such studies are crucial for advancing the understanding of gut microbiota and developing tailored nutrition strategies. Additionally, investigate the effects of peri-workout sugar intake on oral health and microbiota, and its implications for long-term health.Multi-omics integration: Employ comprehensive multi-omics approaches to investigate the direct effects of dietary interventions on recovery and performance, accounting for individual metabolic differences.Nutrigenetics: Validate the impact of genetic variations on the effectiveness of nutritional interventions, especially supplements, on sports performance in larger, independent cohorts of athletes.

## 5. Conclusions

Recent advancements in the application of systems biology to tailor dietary recommendations for endurance athletes have supported and extended established sports nutrition guidelines. The integration of ‘omics’ and wearables technologies presents an opportunity to refine sports nutrition strategies based on athletes’ individual biological profiles. While the evidence on the utility and applicability of most data-driven insights remains preliminary, some hold the potential to enhance recovery, performance, and health. However, the lack of established causal relationships between detected biomarkers and relevant outcomes, along with the inter-individual variability observed, remains a considerable challenge for precision sports nutrition. Future research should prioritize well-powered, replicated crossover RCTs; multivariate N-of-1 clinical trials; comprehensive systems-wide approaches; and the validation of genetic impacts on nutritional interventions to further refine dietary guidelines.

## Figures and Tables

**Figure 1 nutrients-16-03943-f001:**
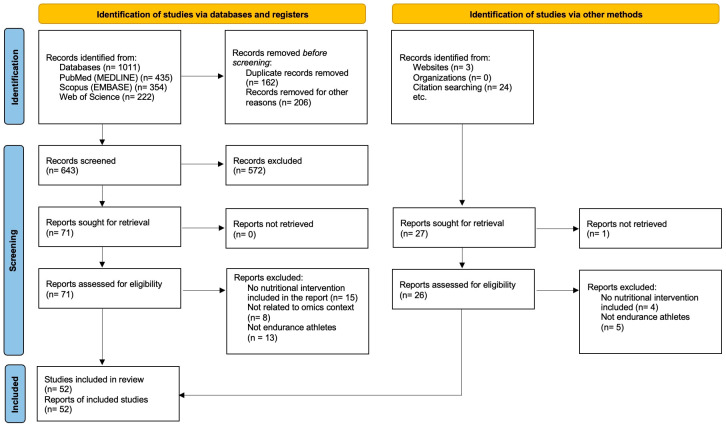
PRISMA flow diagram [25].

**Figure 2 nutrients-16-03943-f002:**
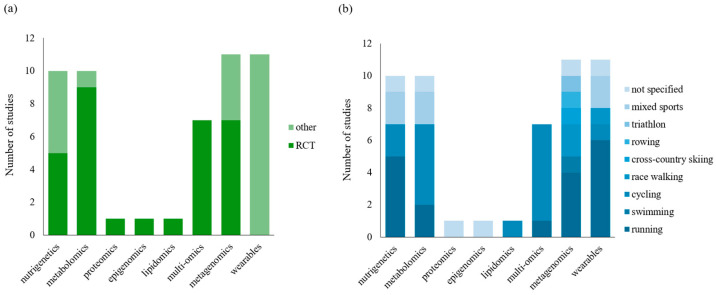
Number of studies identified across individual ‘omics’ or wearables platforms, stratified by (**a**) randomized controlled trials (RCTs) versus other study designs; (**b**) type of endurance sports studied.

## Data Availability

All data available upon request.

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
