# Peer review of "Towards Precision Sports Nutrition for Endurance Athletes: A Scoping Review of Application of Omics and Wearables Technologies"

_nutrients, 2024, doi:10.3390/nu16223943_

Round 1
Reviewer 1 Report
Comments and Suggestions for Authors
Dear editor and authors. Thank you for the opportunity to review this manuscript. I believe a scoping review connecting on sports nutrition, with systems biology in the endurance exercise context could be very useful. However, in the way it is, the review is a little inconclusive and not supporting the scientific advance in the field. The issue is in the core of the question/scope the is not well connected with the search and the results. For example, the selection is only for papers using Omics or wearable technology, but the authors ended up suggesting nutritional recommendations for athletes that goes much beyond the scope, without select all the studies that could generate evidence for the recommendations. In other words, there is a small selection of studies used to generate general recommendations when the initial proposal was the use of systems biology to personalized nutrition. Furthermore, the personalized nutrition is very poorly connected with the rest and become just a subjective opinion along the text with not much evidence of why exactly it should be advantageous if most studies can not assess it. A lot of this could be just that I misunderstood, but in any case, the authors would need to restructure it from the beginning, to deliver a clear, informative and useful review in this field. Therefore, I do not recommend publication of this review.
I found a variety of issues that I described below in case it helps the authors for future submission.
Although the language used in the text is very clear, there are a few aspects of the review that seems to be not clearly defined by the authors and end up confusing the readers. I would recommend making the sessions more coherent and consistent with each Other. For example, (1) If the aim includes wearable Technologies, it needs to be in the title; (2) if the title includes Endurance athletes, it needs to be in the primary objectives of the introduction.
It seems very risky that general recommendations are proposed in a review about precision nutrition. I found the review confusing, maybe because the authors were not sure what questions they need to answer. It seems to me the manuscript is more a critique to current methods on sports nutrition, since there is a lot of observations of how much the methods are good or not, and promises that this will be useful (without robust evidence), but not really many useful “take away” messages. Maybe they are there, but in the way it is written it is not clear.
I tried to understand a bit more of the am by looking to the search descriptors, and it is making a little more confusing since many terms used as synonyms in the search are within each other. For example, the personalized nutrition and personalised sports nutrition is within nutrition, and if you are searching for nutrition, you are getting many papers that do not use personalized or precision nutrition. The same applies to athlete. It did not need to be an endurance athlete and the descriptor athlete encompass all athletes. In regarding to systems biology key word, I understood papers with only one omic analysis approach were included, so it need to be more clear in the aims. I think I understood the idea of the authors accepting wearable technologies, instead of omics but, I think the authors need to be more critical and honest about it. Do the papers included with wearable technology, really made the job we needed? Or they were just the type of assessment in continuous glucose monitoring? Again, it is not clear why continuous glucose monitoring replace system biology I the rational of the study.
About the discursive content of the text, there was too Much generalization and not many specific examples along the text. I understand the topic is broad and it might be challenging. However, when the authors write, for example, about the substantial variability in response to exercise and nutrition, what exactly they want to mean? There are many exercise and nutrition responses!
The manuscript would benefit for a clearer definition of systems biology.
For example, if the second paragraph of introduction can be assumed as the definition of the authors for systems biology, I would say most Papers included in the review can not be used, since they are not considering “multiple” high-throughput omics.
I would recommend the authors to strictly follow PRISMA guidelines, to avoid terminology mistakes than can impair the reader understanding. For example:
-The review was not conduced by prisma guidelines since it is not a guideline to conduce reviews;
-this is not a systematic synthesis of systems biology approaches; this is a scoping review selecting studies via a systematic review and since there are only omics or CGM studies, I understand it is not also systems biology necessarily.
-a scoping review was not done across PubMed, Scopus, and Web of Science; the Search for the scoping review was done in this databses.
-it is not only the data extraction that should be done in a systematic way, but the whole systematic review, and this is more defined by the includion criteria than the methodological approachs during Search, screening, extraction, etc.;
When describing includion criteria, it is importante to explain the PICOS or in this case the PCC clearly. IN the way it is in the abstract, it is very difficult to gues what was reviewed, and therefore the conclusions do not make sense . for example:
The authors said “ Significant inter-individual variability in responses to exercise and nutritional interventions was observed. Furthermore, many studies were constrained by small sample sizes and inadequate reporting of training status.” But if no outcome wwas reported, why this limitation need to appear in the abstract?
It is concluded: “ While evidence is limited, some of the reviewed papers suggest a potential utility of systems biology-driven nutritional recommendations for endurance athletes.” But this utility is never mentioned specifically in the text.
How future studies need t be well-powered replicated for something that should be different for Every person? There are mny conflicts of terminology, with incomplete explanation, like that across the text
Other issues about the search.
Using “Human” filters in the search lead to bias, since many studies are not indexed with a mesh term of human or other animals but they still including one of them. Searching today 14.10.24, I found total of 423 on PubMed (MEDLINE), 294 combining humans and other animals. Therefore, excluding nearly 30 unclassified ones. This is a lot considering 206 have been retrieved in this database.
why reports of websites were included? were they peer reviewed publications? why the 124 studies did not come up with the serach proposed? it suggests the the search was not comprehensive enough.
Reviewer 2 Report
Comments and Suggestions for Authors
In conducting the scoping review, the authors make reference to athletes engaged in endurance sports. However, the specific differences between these athletes are not clearly delineated in this review. It is evident that there are notable differences between the characteristics of rowing athletes and those of marathon athletes, who in turn exhibit distinct qualities when compared to ultramarathon athletes.
It would be beneficial for the authors to group athletes with similar characteristics.
The authors do not differentiate between sex and age, which introduces a clear bias in the comparison of the data.
The authors employ a VO2 value of 65 mL kg-1 min-1 to delineate elite athletes, yet the majority of the studies analysed lack a VO2 value or exhibit a value below this threshold on multiple occasions.
To enhance the completeness of this scoping review, it would be prudent for the authors to prioritise studies that are randomised clinical trials and conduct an analysis of their methodological quality, potentially utilising the PRISMA or PEDro scale.
Furthermore, clarification is needed regarding the correspondence between the results mentioned as 'foundational recommendations' and the references cited, specifically whether they align with 3, 4 and 85.
Additionally, reference 4 is outdated and requires updating by the authors.
Round 2
Reviewer 2 Report
Comments and Suggestions for Authors
All changes and modifications requested from the authors have been made by the authors.
Author Response
Comment: All changes and modifications requested from the authors have been made by the authors.
Response: Thank you for your positive feedback. We appreciate your comments and are pleased that the revisions fulfill all requested modifications.